

# Influence of station density and multi-constellation

# GNSS observations on troposphere tomography

Qingzhi Zhao[1], Kefei Zhang [2,3] and Wanqiang Yao[1]

[1]College of Geomatics, Xi'an University of Science and Technology, Xi'an, China.

[2]School of Environment Science and Spatial Informatics, China University of Mining and

Technology, Xuzhou, China

[3]Satellite Positioning for Atmosphere, Climate and Environment (SPACE) Research Centre, RMIT

University, Melbourne, Australia

**Abstract:** Troposphere tomography, using multi-constellation GNSS observations, has become a
novel approach for the three-dimensional (3-d) reconstruction of water vapour fields. An analysis
of the integration of four Global Navigation Satellite Systems (BeiDou, GPS, GLONASS and
Galileo) observations is presented to investigate the impact of station density and
single/multi-constellation GNSS observations on troposphere tomography. Additionally, the
optimal horizontal resolution of research area is determined in Hong Kong, which considers both
the number of voxels divided, and the coverage rate of discretized voxels penetrated by satellite
signals. Tomography experiment reveals that the influence of station density in a GNSS network is
more significant than the multi-constellation GNSS observations on the reconstruction of 3-d
atmospheric water vapour profiles. Compared to the tomographic result from the
multi-constellation GNSS (BeiDou, GPS, GLONASS and Galileo) observations, the RMS of
SWD residuals derived from the single-GNSS observations has been decreased by 16% when the
data from the other four stations are added. Furthermore, more experiments have been carried out
to analyse the contributions of different combined GNSS data to the reconstructed results, and the
comparisons show some interesting results: (1) the number of iterations used in determining the
weighting matrices of different equations in tomography modelling can be decreased when
considering multi-constellation GNSS observations; (2) the tomographic result with
multi-constellation GNSS data can improve the reconstructed quality of 3-d atmospheric water
vapour by the largest RMS value of about 11% when compared to the PPP-estimated SWD, but
this was not as high as was expected.
**Keywords:** Tropospheric tomography; Multi-constellation GNSS; Station density; Atmospheric
water vapour.

## 1. Introduction

For some years, GNSS-based tropospheric tomography has been regarded as one of the most
promising techniques with which to reconstruct the temporal-spatial variation of atmospheric
water vapour (Flores et al., 2000; Grespi et al., 2008). By discretising the area of interest into
some voxels in different directions, the water vapour information in divided voxels can be



reconstructed with assumption that the unknown estimated parameters are constant during a given
period (Radon, 1917; Flores et al., 2000). So far, this technique has been proved by some
feasibility studies with GPS-only observations (Troller, 2002; Bender and Raabe, 2007; Chen and
Liu, 2014) as well as the simulated multi-constellation GNSS observation (Grespi et al., 2008;
Bender et al., 2011; Wang et al., 2014; Benevides et al., 2015c; Benevides et al., 2017). In addition,
the experimental result reveals that, compared to GPS-only observations, a greater improvement in
the accuracy of tomographic water vapour information using the multi-constellation GNSS
observation has been obtained (Bender et al., 2011; Benevides et al., 2015c; Benevides et al.,
46   2017).

Due to the specific distribution of satellite signals, and the immovability of ground-based stations
in regional network, the geometry of the observed-signal distribution is similar to an inverted cone,
which has a negative effect on tropospheric tomography (Benevides et al., 2015a, 2015b). The
main disadvantage caused by such phenomenon is the sparse filling of the discretised voxels at the
edge and lower sections of the area of interest (Bender and Raabe, 2007). Optimising the design
matrix of observation equation is a way to overcome such bad condition by selecting a
non-uniform symmetrical division of horizontal voxels and a non-uniform thicknesses of the
vertical voxel layers (Nilsson and Gradinarsky, 2006; Yao and Zhao, 2016a, 2016b). Imposing the
satellite rays which come out from the side of the research area onto the reconstructed modelling
is another effective way in which to optimise the structure of the design matrix (Yao and Zhao,
2016b; Yao et al., 2016; Zhao and Yao, 2017). In addition, using more slant-path observations
derived from the upcoming fully-operational GNSS constellations (BeiDou, GLONASS, and
Galileo) is a possible way of solving such issue (Grespi et al., 2008; Bender et al., 2011;
Benevides et al., 2017). Finally, increasing the density of the GNSS network also is a feasible way
to improve the stability and structure of the design matrix (Nilsson and Gradinarsky, 2006).
In most past studies, multi-constellation GNSS observations are simulated with ideal data which
cannot reflect the real conditions of multi-constellation GNSS observations, including the
variations in latitudes, areas, topography, and the surroundings of GNSS stations (Nilsson and
Gradinarsky, 2006; Grespi et al., 2008; Wang et al., 2014). Therefore, the preliminary result
concluded from those studies needs further verification based on the observed multi-constellation
GNSS data, which becomes the focus of this study. In this paper, a method is proposed to
determine the optimal division of voxels in horizontal direction automatically according to the
range of the tomography area as well as the number and distribution of GNSS stations. The
influence of number of stations in a network on the tomographic result is then compared with the
reconstructed wet refractivity field derived from multi-constellation GNSS observations. Finally,
the quality and reliability of tomographic atmospheric water vapour obtained from the different
combined multi-constellation GNSS observations is analysed.
The aim of this research is to analyse the importance and influence of station density and
single/multi-constellation GNSS observations on tropospheric tomography in an upcoming future
scenario of having the multi-constellation GNSS (GPS, BeiDou, GLONASS, and Galileo)
constellations fully operational. The structure of this paper is organised as follows: Sect. II
presents the theory of tropospheric tomography, Sect. III describes the experimental data and the
determination of horizontal resolution. The importance and influence of station density and
single/multi-constellation GNSS observations on troposphere tomography are detailed analysed
and compared in Sects IV and V, respectively, and key conclusions are presented in Sect. VI.






## 2. GNSS tropospheric tomography

Generally, slant wet delay (SWD) and slant water vapour (SWV) are two types of input
observations used in building the observation equations, and the corresponding output results are
wet refractivity and water vapour density, respectively (Flores et al., 2000; Skone and Hoyle, 2005;
Notarpietro et al., 2011; Champollion et al., 2005). Two kinds of reconstructed output information
can be inter-converted with atmospheric temperature field information (Bender et al., 2011). In
this paper, the SWD is selected to reconstruct the atmospheric wet refractivity field.
The zenith tropospheric delay (ZTD) is estimated with high precision using the GNSS observation,
consists of two parts, which includes zenith wet delay (ZWD) and zenith hydrostatic delay (ZHD).
The former can be accurately estimated based on the empirical model, *e.g*., Saastamoinen (1973),
with the observed surface pressure information. Therefore, the latter is obtained by subtracting the
ZHD from ZTD. In our study, the observed multi-constellation GNSS data are processed using the
multi-constellation GNSS Precise Point Positioning (PPP) software with precise orbit and clock
error products (Zhao et al., 2018). Consequently, the SWD can be expressed as:
$$\mathrm{SWD}_{azi,ele} = m_w(ele) \cdot \mathrm{ZWD} + m_w(ele) \cdot cot(ele) \cdot (G_{NS}^w \cdot cos(azi) + G_{WE}^w \cdot sin(azi)) \qquad (1)$$
Where $m_w$ is the wet mapping function. In our processing, the wet Vienna Mapping Function
(VMF) is adopted; $ele$ refers to the satellite elevation angle while $azi$ represents the azimuth
angle. $G_{NS}^w$ and $G_{WE}^w$ are the north-south and west-east gradients of wet delay, respectively,
which are caused by the non-isotropic nature of atmospheric water vapour distributions (Bi et al.,
102 2006).

The SWD value from the satellite to GNSS station antenna is an integral expression, given by:
$$\mathrm{SWD} = 10^{-6} \cdot \int N_w(s) ds \qquad (2)$$
Where $N_w$ represents the wet refractivity (mm/km) and $s$ is the distance over which the
satellite signal penetrates the troposphere (km). According to this tomographic technique, the area
of interest is divided into a number of voxels and the wet refractivity parameters are considered
unchanged during the selected period. Consequently, the total SWD value can be expressed as the
sum of discretised delay parts in each voxel along the satellite ray path, and a linear expression
can be listed as:
$$\mathrm{SWD} = \sum_{i=1}^{m} \sum_{j=1}^{n} \sum_{k=1}^{p} (a_{ijk} \cdot x_{ijk}) \qquad (3)$$
Where $m$ and $n$ are the total number of voxels divided in longitudinal and latitudinal
directions while $p$ is the total number in vertical direction, respectively; $a_{ijk}$ is the distance of
satellite rays, and $x_{ijk}$ is the unknown wet refractivity parameters in voxel $(i, j, k)$, respectively.
Therefore, the observation equation of tomography modelling can be established for all GNSS
stations in a network of interesting area.




As mentioned above, the geometric distribution of satellite rays in the tomographic area is an
inverted cone, thus the design matrix of observation equations is a sparse matrix and not all of the
unknown wet refractivity values are estimated. To solve the problem of rank deficiency, some
external constraints are required (Flores et al., 2000; Troller et al., 2006; Rohm and Bosy., 2011).
Two constraints are imposed in this paper, the one is horizontal weighted constraint, and the other
is the vertical constraint based on the observed radiosonde data in the first three days of the
reconstructed epoch. Consequently, the tomographic modelling imposed the following constraint
equations:
$$\begin{pmatrix} \boldsymbol{A} \\ \boldsymbol{H} \\ \boldsymbol{V} \end{pmatrix} \cdot \boldsymbol{x} = \begin{pmatrix} \boldsymbol{y}_{swd} \\ \boldsymbol{0} \\ \boldsymbol{y}_{rs} \end{pmatrix} \tag{4}$$

Where $\boldsymbol{H}$ represents to the horizontal coefficient matrices while $\boldsymbol{V}$ refers to the vertical
coefficient matrices, respectively. $\boldsymbol{y}_{swd}$ is a vector with SWD values while $\boldsymbol{y}_{rs}$ is the *a priori*
information obtained from the radiosonde information. The form of solution of the unknown wet
refractivity vector can be written as:
$$\hat{\boldsymbol{x}} = (\boldsymbol{A}^T \cdot \boldsymbol{P}_A \cdot \boldsymbol{A} + \boldsymbol{H}^T \cdot \boldsymbol{P}_H \cdot \boldsymbol{H} + \boldsymbol{V}^T \cdot \boldsymbol{P}_V \cdot \boldsymbol{V})^{-1} \cdot (\boldsymbol{A}^T \cdot \boldsymbol{P}_A \cdot \boldsymbol{y}_{swd} + \boldsymbol{V}^T \cdot \boldsymbol{P}_V \cdot \boldsymbol{y}_{rs}) \tag{5}$$

Where $\boldsymbol{P}_A, \boldsymbol{P}_H$ , and $\boldsymbol{P}_V$ are the weighting matrices of observation, horizontal and vertical
equation, respectively. The weighting matrices for different equations are determined by an
optimal weighting method and the homogeneity test was adopted to verify the statistically equality
of three kinds of *a posteriori* unit weight variances (Bartlett, 1937; Guo et al., 2016).

## 3. Tomography experiment and description

### 3.1 Experimental data

A network consisting of fourteen GNSS Satellite Reference Stations (SatRef) in Hong Kong was
selected to perform the tomography experiment during the period of Doy 4 to 26, 2017. The
geographic locations of GNSS and radiosonde stations are presented in Fig. 1. The sampling
interval of the GNSS observations used here was 30 s. The radiosonde station in the experimental
area is used to test the reconstructed result of GNSS troposphere tomography. The range of
tomographic region is from 113.87 °E to 114.35 °E and 22.18 °N to 22.54 °N while the vertical
height is from 0 to 9 km. The horizontal resolution, in voxel terms, is 4 × 12 in latitudinal and
longitudinal directions as determined by an optimal voxel division method, which will be
described below. The vertical resolution adopts a non-uniform vertical layer strategy (Yao and
Zhao, 2016b) with two layers of a thickness of 500 m, three layers of 600 m, four layers of 800 m,
and three layers of 1000 m from the ground to the top of tomography region.



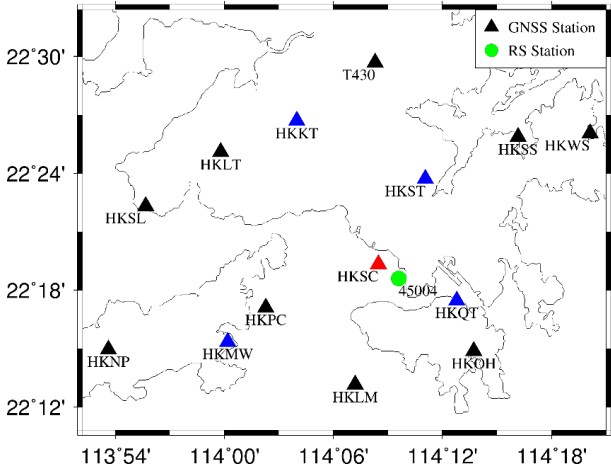

Fig. 1. Geographic location of GNSS and radiosonde stations in SatRef of Hong Kong. The blue
triangles are used to increase the station density, while the station HKSC marked in red and
radiosonde station 45004 marked in green are used to evaluate the performance of tomographic
result

**3.2 Determination of horizontal resolution**
In the procedure of horizontal voxel division, an approach is developed which able to determine
adaptively the optimal horizontal resolution according to the scope of tomography region as well
as the number and distribution of GNSS stations. The specific principle is such that: increasing the
coverage rate of voxels penetrated by satellite signals and optimising the design matrix of the
observation equation, while considering a higher horizontal resolution to reflect the atmospheric
water vapour distribution in as much detail as possible, therefore, a comparative experiment is
performed to validate the developed approach of determining horizontal resolution. Nine schemes
are designed (Table 1): the number of voxels for the bottom layers and the coverage rate of
distributed stations located at the bottom layer are calculated. It can be concluded that Scheme 3
was optimal while considering both the number of voxels divided and the coverage rate of GNSS
stations located in the bottom layers.

168          Table 1. Statistical result of determining horizontal resolution for nine schemes

| Scheme | Longitude ×Latitude | Total voxels | Step of longitude | Step of latitude | Coverage rate of stations (%) |
|---|---|---|---|---|---|
| 1 | 12×9 | 108 | 0.04 | 0.04 | 13.0 |
| 2 | 12×6 | 72 | 0.04 | 0.06 | 18.1 |
| 3 | 12×4 | 48 | 0.04 | 0.09 | 29.2 |
| 4 | 8×9 | 72 | 0.06 | 0.04 | 19.4 |
| 5 | 8×6 | 48 | 0.06 | 0.06 | 25.0 |
| 6 | 8×4 | 32 | 0.06 | 0.09 | 43.8 |
| 7 | 6×9 | 54 | 0.08 | 0.04 | 25.9 |
| 8 | 6×6 | 36 | 0.08 | 0.06 | 36.1 |
| 9 | 6×4 | 24 | 0.08 | 0.09 | 58.3 |




In addition, the coverage rate of the satellite rays for the entire research region is analysed for the
date of doy 4, 2017 under nine combined multi-constellation GNSS observations. In this study, the
time period for each tomography is selected as five minutes. The specific statistical result is
presented in Table 2, where G/C/R/E refer to GPS, BeiDou, GLONASS, and Galileo, respectively.
The conclusion can be drawn that the coverage rate of satellite rays in Schemes 3, 6, 8, and 9 are
relatively large. Considering the number of voxels and coverage rate of stations located in the
bottom layers, Scheme 3 is also considered as the optimal choice. Apart from the above
conclusion, it also can be concluded that the coverage rate of voxels penetrated by satellite signals
for the entire region using two/three/four-GNSS observations are both increased with the
minimum coverage rate by about 5%, when compared to the single-GNSS conditions.
Table 2. Coverage rate of satellite rays for nine combined multi-constellation GNSS observations
(Unit: %)

| Scheme | 1 | 2 | 3 | 4 | 5 | 6 | 7 | 8 | 9 |
|--------|------|------|------|------|------|------|------|------|------|
| G | 51.3 | 60.8 | 72.7 | 61.0 | 69.8 | 81.4 | 67.2 | 76.0 | 85.8 |
| C | 50.0 | 61.2 | 73.9 | 57.4 | 68.5 | 80.6 | 62.2 | 72.6 | 82.5 |
| R | 44.0 | 54.4 | 67.7 | 53.5 | 62.9 | 78.0 | 61.5 | 71.5 | 84.1 |
| E | 30.9 | 40.3 | 53.1 | 40.0 | 50.6 | 64.9 | 47.0 | 57.7 | 72.1 |
| GC | 62.1 | 71.2 | 79.3 | 69.0 | 77.6 | 85.0 | 72.8 | 81.2 | 87.8 |
| GR | 60.4 | 68.8 | 79.5 | 68.0 | 75.8 | 85.2 | 73.1 | 80.9 | 88.5 |
| CR | 59.2 | 69.5 | 79.1 | 65.9 | 75.9 | 84.4 | 70.9 | 80.3 | 86.9 |
| GCR | 65.6 | 74.1 | 81.7 | 71.6 | 80.0 | 86.5 | 75.5 | 83.3 | 89.2 |
| GCRE | 66.9 | 75.3 | 82.3 | 72.5 | 80.5 | 86.8 | 76.1 | 83.6 | 89.5 |


## 4. Importance and influence of station density and multi-constellation GNSS observations on tropospheric tomography

In this section, two schemes are designed to analyse the importance and influence of station
density and multi-constellation GNSS data on the reconstructed atmospheric wet refractivity. For
Scheme 1, all fourteen GNSS stations, as presented by triangles of different colour in Figure 1, are
selected for this tomographic experiment but only considering single-GNSS observations (GPS,
BeiDou, GLONASS, and Galileo, respectively), which are abbreviated to G-14, C-14, R-14, and
E-14, respectively, and the 14 refers to the number of stations used for tomography. For Scheme 2,
only ten GNSS stations are used, as shown by the nine black triangles and one red triangle in
Figure 1, but considering the different multi-constellation GNSS combinations, those
combinations are abbreviated to GC-10, GR-10, CR-10, GCR-10, and GCRE-10, respectively. The
following analysis is focussed on: (1) the investigating of two schemes in the number of GNSS
rays used and coverage rate of the voxels penetrated by GNSS rays, respectively; (2) the
comparison of reconstructed result with radiosonde data as well as the PPP-estimated SWD values
of stations HKSC, respectively.

**4.1 Comparison of GNSS rays used and the coverage rate of voxels penetrated**

23 days of data during the period doy 4-26, 2017 are analysed and the Table 3 shows the mean





value of GNSS rays used and coverage rate of voxels penetrated by signals for the test period. It
can be concluded from the statistical results (Table 3) that the number of signals used in Scheme 2
is apparently large (doubled to tripled) compared to that of Scheme 1, however, the percentage
difference of voxels crossed by rays between Schemes 2 and 1 is not evident expect for the case of
E-14. The number of Galileo satellite observations is small during the test period, therefore, a low
number of signals used and a low coverage rate of voxels penetrated by GNSS signals existed for
the case of E-14 in Scheme 1.
Table 3. Number of GNSS rays used and the coverage rate of crossed voxels in different schemes
during the experimental period

| | Scheme 1 | | | | Scheme 2 | | | | |
|---|---|---|---|---|---|---|---|---|---|
| | G-14 | C-14 | R-14 | E-14 | GC-10 | GR-10 | CR-10 | GCR-10 | GCRE-10 |
| Number of signals used | 974 | 1123 | 693 | 349 | 1433 | 1144 | 1232 | 1905 | 2137 |
| Coverage rate of voxels (%) | 75.3 | 71.8 | 68.0 | 50.0 | 73.8 | 73.6 | 71.2 | 76.9 | 77.4 |

*-14 refers to the statistical result with single-GNSS observations derived from fourteen stations
*-10 refers to the statistical result with multi-constellation GNSS observations derived from ten
stations

To analyse the number of SWDs used and the coverage rate of voxels, the average values of two
schemes for each day is calculated in Figures 2 and 3, respectively. Figure 2 reveals that the
signals used for each day in Scheme 2 is more than double that in Scheme 1: however, Figure 3
reveals that the proportion of voxels penetrated by GNSS signals in Scheme 2 is only about 8%
more than that in Scheme 1.
One point should be noted is that the number of Galileo satellite is lower, therefore, we
re-analysed the SWD numbers and the coverage rate of voxels after removing the case of E-14 in
Scheme 1 (redesignated as Scheme 3). Figures 4 and 5 show the number of SWDS as well as the
proportion of voxels penetrated by GNSS signals without considering the Galileo satellites. From
Figures 4 and 5 we can conclude that the number of signals used in Scheme 2 remains the greatest
at about double that of Scheme 3, but the percentage difference in number of voxels decreased and
only about 3% more than that in Scheme 3. Table 4 lists statistical results relating to SWD
numbers and the coverage rate of voxels for three Schemes mentioned above. From Table 4 we
concluded that, compared to the single-GNSS observations derived from fourteen stations, the
percentage of voxels crossed by rays from the multi-constellation GNSS observations of ten
stations is only increased by 2.9%. Although multiple GNSS observations have been used in
Scheme 2, the coverage rate of voxels did not improve when four stations were removed
compared to that of Scheme 1. This reveals that the station density has a more important influence
on the coverage rate of voxels crossed by rays than multi-constellation GNSS observations.





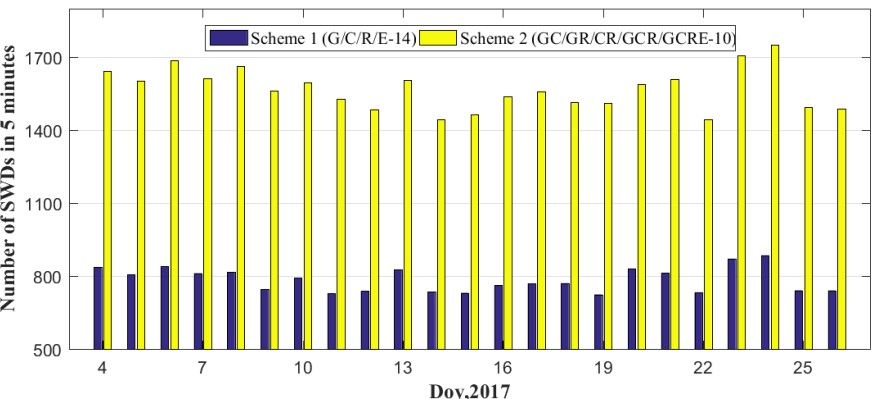


Figure 2. Average number of SWDs used in 5 minutes for two Schemes during the experimental

236                                period

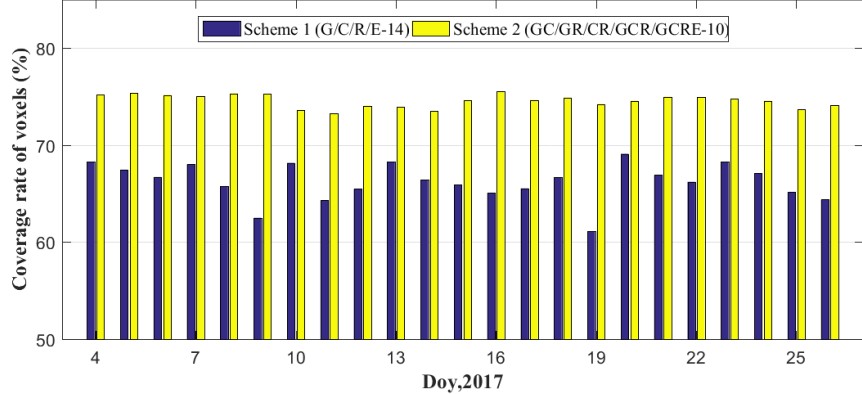


Figure 3. Average coverage rate of voxels penetrated by GNSS signals for two Schemes during the

239                        experimental period

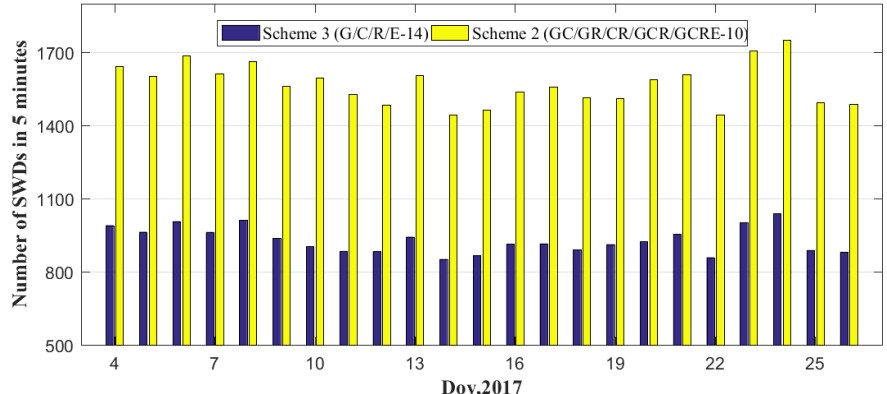


241            Figure 4. Average number of SWDs used in 5 minutes for Schemes 2 and 3 during the



experimental period

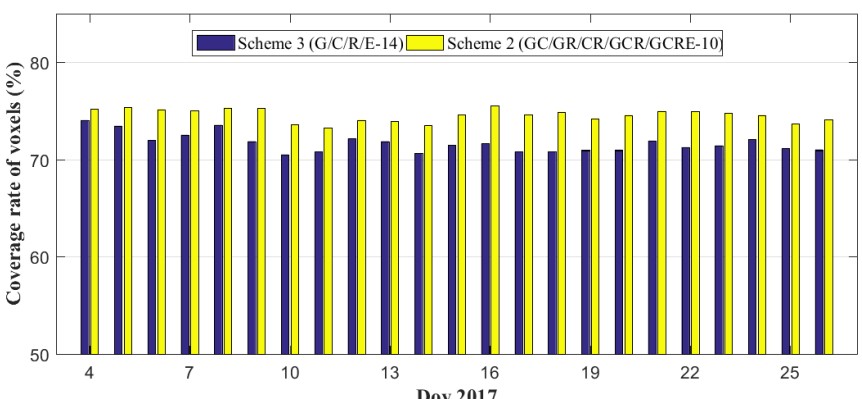


Figure 5. Average coverage rate of voxels penetrated by GNSS signals for Schemes 2 and 3 during
the experimental period

Table 4. Statistical information of GNSS signals used and the percentage of voxels penetrated
during the tested period

| Scheme | Number of signals used | Percentage of crossed voxels (%) |
|---|---|---|
| 1 | 785 | 66.2 |
| 2 | 1570 | 74.6 |
| 3 | 930 | 71.7 |


## 4.2 Comparison with radiosonde data

In this section, we further compared the influence of station density on the tomographic result. In
the experimental area, there is a radiosonde station, as shown by the green circle in Figure 1.
Several studies have proved that radiosonde data has a high accuracy in providing the water
vapour profiles (Niell et al., 2001; Liu et al., 2013), and the result calculated from radiosonde is
used as a reference in this paper to evaluate the tomographic result. The comparison experiment of
reconstructed wet refractivity profile information of different Schemes at the radiosonde station
with the radiosonde data is carried out at two specific epochs (UTC 00:00 and 12:00, respectively).
Figure 6 shows the root mean square (RMS) error of wet refractivity difference between different
tomography conditions and radiosonde data. Table 5 gives the specific statistical information
pertaining to RMS, bias, and mean absolute error (MAE) for different Schemes. From Figure 6
and    Table    5,    we    can    conclude    that    the    tomographic    results    using    different
single/multi-constellation GNSS observations are similar at the radiosonde location. As presented
in Figure 1, station HKSC is near the radiosonde station, therefore, the reconstructed atmospheric
wet refractivity from different cases nearby the location of radiosonde station are relatively
accurate and undifferentiated; however, such a result cannot represent the quality of reconstructed
results of wet refractivity fields for the entire region. Therefore, the performance of the
tomographic result for the entire research region is further evaluated using the PPP-estimated
SWDs below.




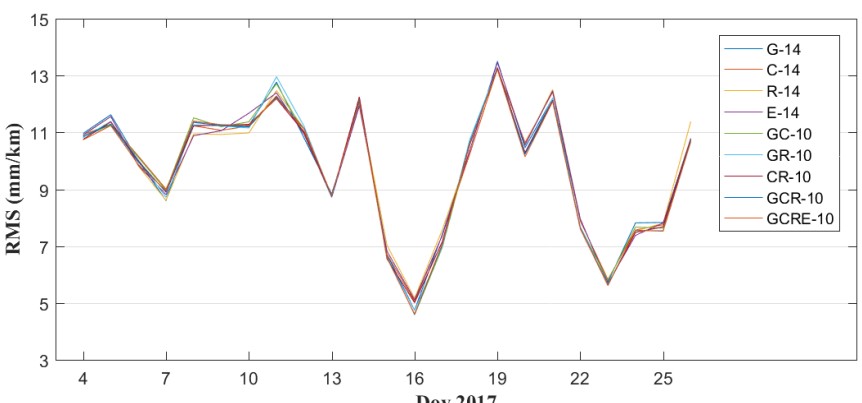


Figure 6. RMS error of wet refractivity difference derived from various conditions during the experiment period

Table 5. Statistical result of RMS, Bias and MAE of wet refractivity difference for different Schemes during the experimental period

| Scheme | | RMS (mm/km) | Bias (mm/km) | MAE (mm/km) |
|---|---|---|---|---|
| 1 | G-14 | 9.78 | 1.54 | 7.12 |
| | C-14 | 9.78 | 1.55 | 7.14 |
| | R-14 | 9.75 | 1.64 | 7.15 |
| | E-14 | 9.76 | 1.66 | 7.14 |
| 2 | GC-10 | 9.72 | 1.40 | 7.10 |
| | GR-10 | 9.71 | 1.40 | 7.10 |
| | CR-10 | 9.72 | 1.46 | 7.10 |
| | GCR-10 | 9.68 | 1.41 | 7.07 |
| | GCRE-10 | 9.66 | 1.42 | 7.07 |

275

## 4.3 Comparison with PPP-estimated SWDs

To assess the reconstructed result of the entire region, two new schemes are designed: Scheme 1, using only the single-GNSS observations from thirteen GNSS stations (except for HKSC) is used for reconstructing the atmospheric wet refractivity; Scheme 2, nine GNSS stations, as shown by the black triangles in Figure 1, are selected using combined multi-constellation GNSS observations. The slant wet delays of station HKSC are computed based on the different tomographic results and their differences against the multi-constellation GNSS PPP-estimated slant wet delays are also obtained. The RMS and MAE of SWD residuals for each day in two schemes are presented in Figures 7 and 8, where the red dashed line represents the average RMS and MAE obtained under conditions G-13, C-13, R-13, and E-13 while the blue dashed line represents the average RMS and MAE obtained from cases GC-9, GR-9, CR-9, GCR-9, and GCRE-9, respectively. Figures 7 and 8 reveal that the average RMS and MAE of Scheme 1 is mostly smaller than that of Scheme 2 over the experimental period, which shows that the reconstructed atmospheric wet refractivity field of Scheme 1 over the entire research area is




superior to the tomographic result of Scheme 2. Statistical results pertaining to different schemes
are listed in Table 6, from which it is seen that, compared to Scheme 2, the average RMS and
MAE accuracy of Scheme 1 is increased by 16% and 33.4%, respectively. Thence it was
concluded that, compared to the tomographic result of multi-constellation GNSS observations,
increasing the station density has greater significance to the reconstruction of the atmospheric
water vapour field.

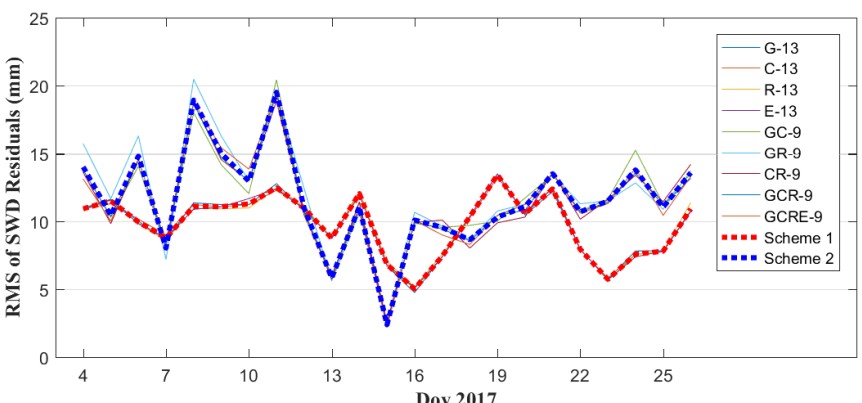

Figure 7. Average RMS of SWD residuals for different schemes over the experimental period

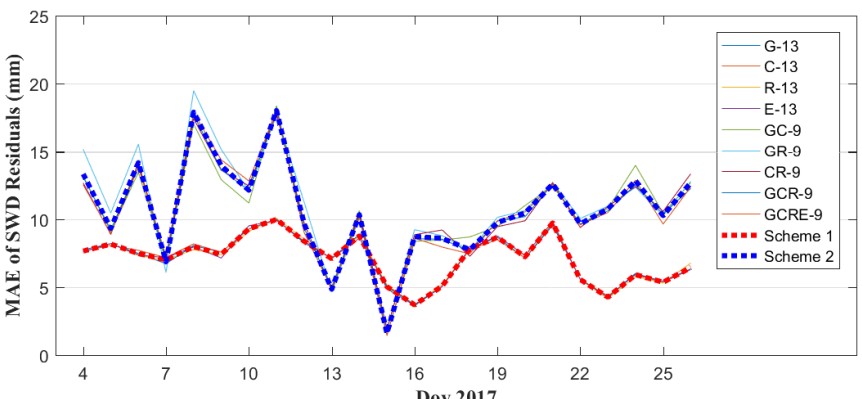

Figure 8. Average RMS of MAE residuals for different schemes over the experimental period
301           Table 6. Statistical result of RMS and MAE for two schemes over the experimental period

| Scheme | | RMS | MAE |
|---|---|---|---|
| 1 | G-14 | 9.78 | 7.12 |
| | C-14 | 9.77 | 7.14 |
| | R-14 | 9.79 | 7.15 |
| | E-14 | 9.76 | 7.14 |
| 2 | GC-10 | 11.64 | 10.62 |
| | GR-10 | 11.99 | 11.09 |
| | CR-10 | 11.50 | 10.66 |



| | GCR-10 | 11.55 | 10.61 |
| | GCRE-10 | 11.52 | 10.58 |

## 5 Analysis of multi-constellation GNSS troposphere tomography

### 5.1 Comparison of signals used and coverage rate of voxels penetrated

Here, all fourteen GNSS stations are selected to reconstruct the atmospheric wet refractivity, and the tomographic results derived from different multi-constellation GNSS observations are compared and analysed. Nine types of single/multi-constellation GNSS observations are designed in schemes designated: G-14, C-14, R-14, E-14, GC-14, GR-14, CR-14, GCR-14, and GCR-14, respectively. Before evaluating the performance of the tomographic result, the average number of GNSS signals used and the percentage of voxels penetrated over the experimental period for each tomography step are first analysed (Table 7). Table 7 reveals that compared to schemes G-14 C-14, R-14, and E-14, multi-constellation GNSS schemes have more voxels crossed by rays, but the change is small with respect to changing SWD numbers.

Table 7. Statistical information of number of GNSS rays used and the coverage rate of voxels penetrated

| | G-14 | C-14 | R-14 | E-14 | GC-14 | GR-14 | CR-14 | GCR-14 | GCRE-14 |
| --- | --- | --- | --- | --- | --- | --- | --- | --- | --- |
| Number of signals used | 974 | 1123 | 693 | 349 | 2097 | 1168 | 1816 | 2791 | 3139 |
| Coverage rate of voxels (%) | 75.3 | 71.8 | 68.0 | 50.0 | 80.0 | 79.8 | 78.8 | 82.0 | 82.4 |

### 5.2 Evaluation of multi-constellation GNSS troposphere tomography

To analyse the performance of the multi-constellation GNSS troposphere tomography, the wet refractivity profile derived from nine schemes is first compared with the result from the radiosonde data thereat. The average RMS, Bias and MAE of wet refractivity difference between different schemes and radiosonde data over the experimental period are calculated (Table 8). As mentioned in Section 2, an iterative produce is required to determine the weighting matrices of different equations in tomographic modelling. Therefore, the number of iterations and the average elevation angle of satellite signals for different schemes are also considered (Table 8). It can be observed from Table 8 that the average RMS, bias, and MAE of different schemes are similar, which reflects the fact that the reconstructed wet refractivity profile obtained from different schemes applied at the radiosonde station have equivalent accuracy.

However, the number of iterations of various schemes are different when determining the weighting matrices of the different types of equations used in tomographic modelling. By analysing the relationship between the number of iterations and elevation angles over the tested period, a negative linear relationship is found between two factors and the fitted data are presented in Figure 9. Such a negative correlation reveals that the resolving time of tomographic modelling can be decreased with multi-constellation GNSS observations, which is important in the real-time reconstruction of atmosphere water vapour data.

Table 8. Statistical result of average RMS, Bias, MAE, elevation angle and iteration times for






| Scheme | RMS | Bias | MAE | Iteration times | Elevation angle (°) |
|--------|-----|------|-----|-----------------|---------------------|
| different schemes over the experimental period | | | | | |
| G-14 | 9.78 | 1.54 | 7.12 | 4.8 | 39.8 |
| C-14 | 9.77 | 1.55 | 7.14 | 3.5 | 51.9 |
| R-14 | 9.79 | 1.64 | 7.15 | 5.0 | 40.2 |
| E-14 | 9.76 | 1.66 | 7.14 | 4.2 | 44.5 |
| GC-14 | 9.76 | 1.54 | 7.11 | 4.1 | 45.8 |
| GR-14 | 9.75 | 1.52 | 7.10 | 5.1 | 40.0 |
| CR-14 | 9.78 | 1.56 | 7.14 | 4.2 | 46.1 |
| GCR-14 | 9.76 | 1.55 | 7.09 | 3.8 | 44.0 |
| GCRE-14 | 9.75 | 1.55 | 7.10 | 3.7 | 44.1 |

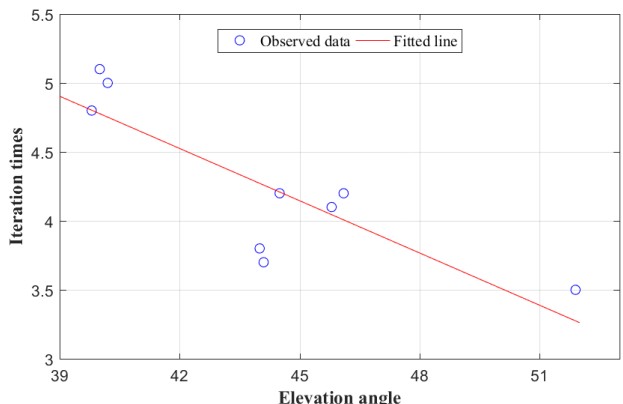


Figure 9. Relationship between iteration times and elevation angle during the experimental period

As mentioned above, the accuracy of different schemes evaluated for the location of radiosonde
cannot represent the tomographic quality across the entire region, therefore, a further comparison
is carried out using only thirteen GNSS stations in the network except for station HKSC. The slant
wet delays of station HKSC, estimated using multi-GNSS PPP software, are compared with the
calculated SWDs derived from different schemes. Figures 10 and 11 show the average RMS and
MAE of SWD residuals on each day during the experiment, where the blue dashed line represents
the average of RMS and MAE obtained from schemes G-13, C-13, R-13, and E-13, while the red
dashed line represents the average of RMS and MAE obtained from schemes GC-13, GR-13,
CR-13, GCR-13, and GCRE-13. From those two Figures it was found that the reconstructed
quality of atmospheric wet refractivity field data for the entire region using multi-constellation
GNSS observations has been improved slightly, when compared to that using single-constellation
GNSS data. By analysing the statistical results pertaining to different schemes (Table 9) it was
found that, compared to the single-constellation GNSS troposphere tomography, RMS accuracy of
the multi-constellation GNSS troposphere tomography improved by about 10%.



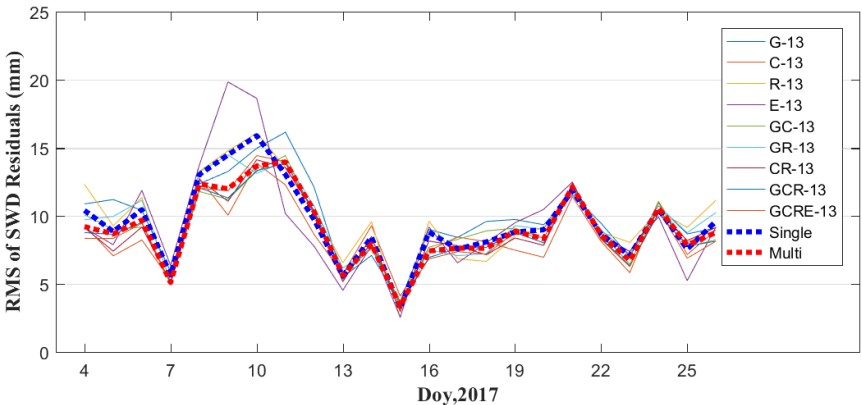


Figure 10. Average RMS of SWD residuals for different schemes over the experimental period

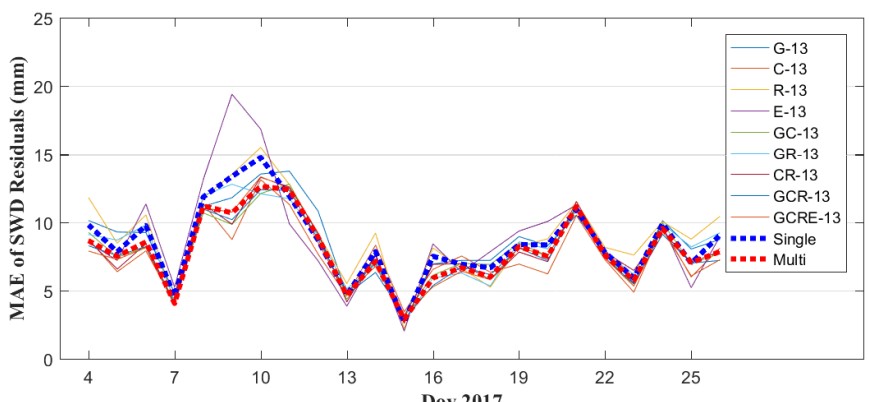


Figure 11. Average MAE of SWD residuals for different schemes over the experimental period
Table 9. Statistical result of RMS, Bias and MAE of SWD residuals from different schemes over
the experimental period

| Scheme | RMS | Bias | MAE |
|--------|-----|------|-----|
| G-13 | 9.83 | 6.71 | 8.62 |
| C-13 | 8.58 | 6.34 | 8.58 |
| R-13 | 9.05 | 7.65 | 9.05 |
| E-13 | 9.41 | 7.62 | 8.83 |
| GC-13 | 9.03 | 6.44 | 7.96 |
| GR-13 | 9.40 | 6.66 | 8.28 |
| CR-13 | 8.89 | 6.78 | 7.96 |
| GCR-13 | 8.78 | 6.38 | 7.77 |
| GCRE-13 | 8.75 | 6.36 | 7.73 |


## 6 Conclusion


The observed multi-constellation GNSS (GPS, BeiDou, GLONASS, and Galileo) observations





have been used to investigate the importance and influence of station density and multi-GNSS
constellation data on troposphere tomography. The SWDs of fourteen GNSS stations in a network
in Hong Kong are estimated using the multi-constellation GNSS PPP software.
For GNSS troposphere tomography, the horizontal resolution of voxels is first determined
according to the number of voxels and the coverage rate of GNSS stations located in the bottom
layers. A comparative experiment using single/multi-constellation GNSS data derived from
different numbers of stations revealed that increasing the station density improved the quality of
tomographic results with the RMS accuracy of SWDs residuals increasing by about 16%, when
compared to the result obtained when using multi-constellation GNSS troposphere tomography. In
addition, compared to the single-constellation GNSS observations, troposphere tomography using
multi-constellation GNSS data can: (1) reduce the resolving time when determining the weighting
matrices of different equations used in tomographic modelling, which has practical significance
for the real-time reconstruction of atmospheric water vapour profiles; and (2) improve the quality
of tomographic results to a certain extent.
With the upcoming full operability of the multi-constellation GNSS, it is expected to increase the
number of SWDs used for troposphere tomography. Although the improvement of reconstructed
results is not as was expected, it was mainly determined by the spatial distribution of GNSS
stations, multi-constellation GNSS troposphere tomography is also worth studying, especially for
potential application of this technique in real-time atmospheric water vapour reconstruction.

**Acknowledgments:** The authors thank IGAR (Integrated Global Radiosonde Archive) for
providing the radiosonde data. The Lands Department of HKSAR and Hong Kong Observatory
are also acknowledged for providing GNSS and the corresponding meteorological data. This
research was funded by the State Key Program of National Natural Science Foundation of China
389  (41730109).


**Conflicts of Interest:** The authors declare no conflict of interest.

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
