# Peer review of "Influence of station density and multi-constellation"

_Annales Geophysicae, 2018_

## Referee Comment (RC1) · Anonymous Referee #1 · 25 Oct 2018

General comments: This manuscript discusses the impact of the number of GNSS stations and the use of single/multiple GNSS constellations on the tomography results. For this purpose, this study conducts a lot of tomography experiments in Hong Kong. This study may have some reference significance, but still has some deficiencies. My major concerns are your experiment designs and key results. I have specified these points and other comments in the specific comments. In addition, the language needs significant improvement. Though I have pointed out some, there are still many other problems.

Specific comments: Lines 62-63: In most past studies, multi-constellation GNSS observations are simulated with ideal data which cannot reflect the real conditions of multi-constellation GNSS observations. Please be more careful to say this and check the recent publications

Line 159-161: The specific principle is such that: increasing the 159 coverage rate of voxels penetrated by satellite signals and optimising the design matrix of the 160 observation equation. This is your criterion to determine the best horizontal division of the voxels. But it is not clear to me how you assess the state of the design matrix. From lines 157-167, I cannot make a sense of what your adaptive method to determine the horizontal division is. I am also not convinced why you choose scheme 3.

Lines 227-231: I don't think the experiment and the statistics in Table 3 support your conclusion since your experiment is poorly designed and the comparison is not fair at all. I am surprised why you design such a comparison rather than single-GNSS (14 sites) vs. multi-GNSS (14 sites) and multi-GNSS (10 sites) vs. multi-GNSS (14 sites)?

Line 263-265: station HKSC is near the radiosonde station, therefore, the reconstructed atmospheric wet refractivity from different cases nearby the location of radiosonde station are relatively accurate and undifferentiated. Is this because that HKSC always has enough observations? Do you use the radiosonde data of the tomographic epoch as the a priori information?

Figures 7 and 8: difficult to distinguish the different lines. Try to use more differentiable color.

Table 8: the presented results surprised me. The all-GNSS scheme does not even outperform the Galileo-only scheme! I also don't think the close distance between the radiosonde station and the HKSC station can explain the negligible RMSE differences among the 9 schemes. Again, is it due to that you use the radiosonde of the tomography epoch as the a priori values?

Lines 15-16: the expression is very confusing, please be specific and accurate. Lines

17-19: the expression is too general and inaccurate, please be specific. Try to revise it to something like "The results show that densification of the GNSS network plays a more important role than using multi-constellation GNSS observations in improving the retrieval of . . . . . .". Lines 19-22: the expression is redundant. "Compared to the tomographic result from the 19 multi-constellation GNSS. . . . . . when the 21 data from the other four stations are added". Line 22: "more" -> "additional" Lines 26-29: unreadable expression Line 35: delete "with which" Line 37: "some" -> "finite" and delete "different directions" Line 39: "proved" -> "proven" Lines 42-45: poor expression Lines 47-49: try to simplify the expression and be accurate. Line 50: what does the "sparse filling" mean? Be specific Lines 51-54: you never talked about "design matrix" and its link with the previously mentioned disadvantage before this expression. Though I can understand you, most readers will get lost here. Try to give a clear logic link. Line 55: "modeling" -> "model" Line 56: delete "in which" Line 59: "way of solving such" -> "way to solve this" Line 60: "increasing the density of the GNSS network. . . . . .also is a . . . . . ." -> "densifying the GNSS network. . . . . .is another. . . . . ." Lines 70-71: these two different things are incomparable Lines 74-77: rephrase this sentence Line 80: "detailed" -> "detailedly" Line 92: "former" -> "latter" Line 93: "the latter" -> "the ZWD" Line 109: delete ", and a linear expression can be listed as", it is redundant Line 118: "not all of the unknown wet refractivity values are estimated" -> "not all of the unknowns can be determined" Line 133: "statistically" -> "statistical" Line 157: delete "which able" Line 159: delete "such" Line 160: specify "coverage rate" Line 188: delete "stations, as presented by triangles of different colour in Figure 1,", redundant Line 200: delete "the" Line 203: "doubled to tripled" -> "double to triple" Line 204-205: R-14 is also evident Line 385: "IGAR" -> "IGRA"

---

## Referee Comment (RC2) · Anonymous Referee #2 · 22 Nov 2018

General comments This paper examines some aspects of tropospheric tomography using GNSS signals via designed experiments. The main purpose is to investigate the impact of station density and multi-constellation systems involved in the process of estimating a better representation of water vapour in space.

Specific comments Did not quite understand the selection of the various schemes. For examples why 10 vs 14 stations? What is the basis for this choice? Do the differences between schemes in terms of RMSE as presented in Table 8 justify the main claim of the paper?

Technical corrections 29 this was not as high as expected. 37 voxels in different direc-

tions 38 reconstructed under the assumption that the unknown 67 GNSS data, which is the focus 68 determine the optimal division of voxels in the horizontal direction 70 influence of the number of stations in a network 72 the quality and reliability of tomographic atmospheric water vapour obtained from different 80 single/multi-constellation GNSS observations on troposphere tomography are analysed in detail 90-94 Wrong usage of former and latter must rephrase 157 In the procedure of horizontal voxel division, an approach is developed which enables the determination 177 Further to the conclusion above it can also be concluded 178 for the entire region using two/three/four-GNSS observations both increase with the 195 following analysis focuses on: (1) investigating of two schemes in 204 difference of voxels crossed by rays between Schemes 2 and 1 is not as expected for the case of 219 It should be noted that the number of Galileo satellite is lower 221 SWDs 222 Galileo satellites 223 the highest 225 only by about 3% more than 226 of voxels for the three Schemes 288 lower than that 292 Hence it was 323 an iterative procedure 379 The upcoming full operability of the multi-constellation GNSS, is expected 381 results is not as expected

---

## Author Comment (AC1) · 11 Dec 2018

The comment was uploaded in the form of a supplement:
https://www.ann-geophys-discuss.net/angeo-2018-106/angeo-2018-106-AC1-supplement.zip

---

## Author Comment (AC2) · 11 Dec 2018

The comment was uploaded in the form of a supplement: https://www.ann-geophys-discuss.net/angeo-2018-106/angeo-2018-106-AC2-supplement.zip

---

## Author Response (AR1)

**Response to the Editor's Comments**

In view of the comments of the referees, I inform you that the submitted paper could be considered for publication in Annales Geophysicae subject to major revisions.

Please revise your manuscript and provide a point-by-point reply to the reviewers' comments.

Thanks for the editor's warm work earnestly, we have revised the manuscript carefully according to the reviewers' comments and suggestions, please see the responses below.

The following is a point-to-point response to the reviewers' comments.

Response to Referee #1:

**General comments**

This manuscript discusses the impact of the number of GNSS stations and the use of single/multiple GNSS constellations on the tomography results. For this purpose, this study conducts a lot of tomography experiments in Hong Kong. This study may have some reference significance, but still has some deficiencies. My major concerns are your experiment designs and key results. I have specified these points and other comments in the specific comments. In addition, the language needs significant improvement. Though I have pointed out some, there are still many other problems.

✓ Thanks for the reviewer's comments and suggestions, all the specific comments and suggestions have been answered point-to-point in the following. In addition, this manuscript has been proofread by a native English speaker.

**Specific comments**

Lines 62-63: In most past studies, multi-constellation GNSS observations are simulated with ideal data which cannot reflect the real conditions of multi-constellation GNSS observations. Please be more careful to say this and check the recent publications

✓ Thanks for the reviewer's reminding, this expression has been revised and the corresponding descriptions of the current situation of GNSS tomography have been added, please see in P2, Lines 61-63 and Lines 66-69.

Line 159-161: The specific principle is such that: increasing the coverage rate of voxels penetrated by satellite signals and optimising the design matrix of the observation equation. This is your criterion to determine the best horizontal division of the voxels. But it is not clear to me how you assess the state of the design matrix.

✓ We are sorry for our improper expression, here, we want to express that the structure of design matrix can be improved by increasing the number of voxels crossed by satellite signal. Thanks for the reviewer's reminding, this expression has been revised in P5 Lines 162-165.

From lines 157-167, I cannot make a sense of what your adaptive method to determine the horizontal division is. I am also not convinced why you choose scheme 3.

✓ Thanks for the reviewer's question, we are sorry for our improper expression, the word 'adaptive' has been deleted. the method to determine the horizontal division is based on the principle, which guarantees the relatively large coverage rate of GNSS stations located in the bottom layer to optimize the design matrix of the observation equation, and considers a higher horizontal resolution to reflect the atmospheric water vapour distribution in as much detail as possible. For most past studies, the horizontal resolution of tomography area is selected according to the experience (e.g. 10 km, 20 km) but didn't give the reason.

✓ In table 1, nine schemes are given to select the horizontal resolution. Scheme 3 is determined according to the total number of divided voxels and the coverage rate of GNSS stations located in the bottom layer. Because the water vapor content is mainly concentrated on the low layers, and the tomographic result is largely affected by the distribution of GNSS observation in the low layers. Therefore, the large coverage rate of GNSS stations in the bottom layer means a large distribution of GNSS observation in the low layers, which is favorable to the final tomographic result.

Lines 227-231: I don't think the experiment and the statistics in Table 4 support your conclusion since your experiment is poorly designed and the comparison is not fair at all. I am surprised why you design such a comparison rather than single-GNSS (14 sites) vs. multi-GNSS (14 sites) and multi-GNSS (10 sites) vs. multi-GNSS (14 sites).

✓ We appreciate for the reviewer's suggestion, we have re-designed the comparison experiment in section 4.1, and four schemes have been designed, which are single-GNSS (10 sites), multi-GNSS (10 sites), single-GNSS (14 sites) and multi-GNSS (14 sites). In addition, all the descriptions and conclusions related to this section have been rewritten, please see in P6-9.

Line 263-265: station HKSC is near the radiosonde station, therefore, the reconstructed atmospheric wet refractivity from different cases nearby the location of radiosonde station are relatively accurate and undifferentiated. Is this because that HKSC always has enough observations? Do you use the radiosonde data of the tomographic epoch as the a priori information?

✓ Thanks for the reviewer's question, in our opinion, HKSC always has observations but we not sure whether it has enough observations.

✓ Yes, the radiosonde data of the tomographic epoch is also used as the a priori information for the location of radiosonde station, which has been described in P4 Lines135-137.

✓ Thanks for the reviewer's reminding, the reasons for the similar tomographic result of different cases have been revised and given in P9 Lines 263-267.

Figures 7 and 8: difficult to distinguish the different lines. Try to use more differentiable color.

✓ Thanks for the reviewer's suggestion, we have tried our best to distinguish the different lines using different colors in Figures 7 and 8, due to the differences between thoses schemes are small, it is very difficult to distinguish them obviously.

Table 8: the presented results surprised me. The all-GNSS scheme does not even outperform the Galileo-only scheme! I also don't think the close distance between the radiosonde station and the HKSC station can explain the negligible RMSE differences among the 9 schemes. Again, is it due to that you use the radiosonde of the tomography epoch as the a priori values?

✓ Yes, we totally agree with the reviewer's opinion that the similar tomographic results obtained for 9 schemes in Table 8 are related to the use of radiosonde data as the priori value for the location of radiosonde station. However, we think this may be also associated with the short distance between radiosonde and HKSC station, therefore, the reasons for the similar tomographic result of different cases have been revised and given in P9 Lines 263-267.

✓ In addition, a further comparison has been performed not only for the location of HKSC but also for the entire tomography area in the following part and the compared results have been presented in Figures 10 and 11 as well as in Table 9, from which it can be observed that the all-GNSS schemes are generally outperform the single-GNSS schemes.

Lines 15-16: the expression is very confusing, please be specific and accurate.

✓ Thanks for the reviewer's reminding, this expression has been revised in P1 Lines 15-17.

Lines.17-19: the expression is too general and inaccurate, please be specific. Try to revise it to something like "The results show that densification of the GNSS network plays a more important role than using multi-constellation GNSS observations in improving the retrieval of : : :: : :".

✓ We appreciate for the reviewer's suggestion; this expression has been revised in P1 Lines 17-19.

Lines 19-22: the expression is redundant. "Compared to the tomographic result from the multi-constellation GNSS: : :: : : when the data from the other four stations are added".

✓ Thanks for the reviewer's suggestion, the redundant content has been deleted.

Line 22: "more" -> "additional"

✓ Thanks for the reviewer's reminding, the word 'more' has been replaced by 'additional'.

Lines 26-29: unreadable expression

✓ Thanks for the reviewer's reminding, this expression has been revised in P1 Lines 24-27.

Line 35: delete "with which"

✓ Thanks for the reviewer's suggestion, the 'with which' has been deleted in the manuscript.

Line 37: "some" -> "finite" and delete "different directions"

✓ Thanks for the reviewer's suggestion, the word 'some' has been replaced by 'finite', and the 'different directions' has been deleted.

Line 39: "proved" -> "proven"

✓ Thanks for the reviewer's suggestion, the word 'proved' has been replaced by 'proven'.

Lines 42-45: poor expression

✓ Thanks for the reviewer's reminding, this expression has been rewritten in P2, Lines 40-42.

Lines 47-49: try to simplify the expression and be accurate.

✓ Thanks for the reviewer's reminding, this expression has been rewritten in P2, Lines 44-46.

Line 50: what does the "sparse filling" mean? Be specific

✓ Thanks for the reviewer's reminding, the description of 'sparse filling' has been given in P2, Lines 48-49.

Lines 51-54: you never talked about "design matrix" and its link with the previously mentioned disadvantage before this expression. Though I can understand you, most readers will get lost here. Try to give a clear logic link.

✓ Thanks for the reviewer's reminding, a logic link has been given using a sentence in P2, Lines 49-50.

Line 55: "modeling" -> "model"

✓ Thanks for the reviewer's reminding, the word 'modeling' has been replaced by 'model'.

Line 56: delete "in which"

✓ We appreciate for the reviewer's suggestion; the 'in which' has been deleted.

Line 59: "way of solving such" -> "way to solve this"

✓ Thanks for the reviewer's suggestion, this expression has been revised.

Line 60: "increasing the density of the GNSS network: : :: : :also is a : : :: : :" -> "densifying the GNSS network: : :: : :is another: : :: : :"

✓ We appreciate for the reviewer's suggestion; this sentence has been revised.

Lines 70-71: these two different things are incomparable

✓ Thanks for the reviewer's reminding, this sentence has been revised in P2 Lines 72-74.

Lines 74-77: rephrase this sentence

✓ Thanks for the reviewer's suggestion, this sentence has been rephrased P2 Lines 77-79.

Line 80: "detailed" -> "detailedly"

✓ Thanks for the reviewer's comments, this expression has been revised.

Line 92: "former" -> "latter"

✓ Thanks for the reviewer's reminding, this expression has been revised.

Line 93: "the latter" -> "the ZWD"

✓ Thanks for the reviewer's reminding, this expression has been revised.

Line 109: delete ", and a linear expression can be listed as", it is redundant

✓ Thanks for the reviewer's reminding, the redundant content has been deleted.

Line 118: "not all of the unknown wet refractivity values are estimated" -> "not all of the unknowns can be determined"

✓ Thanks for the reviewer's reminding, this expression has been revised.

Line 133: "statistically" -> "statistical"

✓ Thanks for the reviewer's reminding, this word has been corrected.

Line 157: delete "which able"

✓ Thanks for the reviewer's suggestion, this sentence has been revised and the word 'which able' has been deleted.

Line 159: delete "such"

✓ Thanks for the reviewer's reminding, the word 'such' has been deleted.

Line 160: specify "coverage rate"

✓ Thanks for the reviewer's reminding, the coverage rate has been specified in P5, Lines 167-168.

Line 188: delete "stations, as presented by triangles of different colour in Figure 1,", redundant

✓ Thanks for the reviewer's reminding, the redundant content has been deleted.

Line 200: delete "the"

✓ Thanks for the reviewer's reminding, the word 'the' has been deleted.

Line 203: "doubled to tripled" -> "double to triple"

✓ Thanks for the reviewer's reminding, this expression has been revised.

Line 204-205: R-14 is also evident

✓ Thanks for the reviewer's comment, this comparison has been re-designed and all the descriptions and conclusions have been rewritten.

Line 385: "IGAR" -> "IGRA".

✓   Thanks for the reviewer's reminding, this expression has been revised.

Response to Referee #2:

**General comments**

This paper examines some aspects of tropospheric tomography using GNSS signals via designed experiments. The main purpose is to investigate the impact of station density and multi-constellation systems involved in the process of estimating a better representation of water vapour in space.

**Specific comments**

Did not quite understand the selection of the various schemes. For examples why 10 vs 14 stations? What is the basis for this choice? Do the differences between schemes in terms of RMSE as presented in Table 8 justify the main claim of the paper?

✓ Thanks for the reviewer's question, the selection of the various schemes in Section 4.1 have been re-designed according to the other reviewer's suggestion. Therefore, the schemes of single-GNSS (10 sites), multi-GNSS (10 sites), single-GNSS (14 sites) and multi-GNSS (14 sites) are determined to better investigate the number of GNSS rays used and coverage rate of the voxels penetrated by GNSS rays under different cases. Additionally, all the descriptions and conclusions related to this section have been rewritten, please see in P6-9.

✓ In our opinion, the differences between schemes in terms of RMSE as presented in Table 8 cannot justify the main claim of the paper completely, therefore, the further comparison of SWDs has been performed in the following part and the corresponding conclusion can be obtained from Figures 9 and 10 as well as Table 9.

**Specific comments**

29 this was not as high as expected.

✓ Thanks for the reviewer's reminding, this expression has been revised.

37 voxels in different directions

✓ Thanks for the reviewer's reminding, this expression has been revised.

38 reconstructed under the assumption that the unknown

✓ Thanks for the reviewer's reminding, this expression has been revised.

67 GNSS data, which is the focus

✓ Thanks for the reviewer's reminding, this expression has been revised.

68 determine the optimal division of voxels in the horizontal direction

✓ Thanks for the reviewer's reminding, this expression has been revised.

70 influence of the number of stations in a network

✓ Thanks for the reviewer's reminding, this expression has been revised.

72 the quality and reliability of tomographic atmospheric water vapour obtained from different
✓    Thanks for the reviewer's reminding, this expression has been revised.

80 single/multi-constellation GNSS observations on troposphere tomography are analysed in detail
✓    Thanks for the reviewer's reminding, this expression has been revised.

90-94 Wrong usage of former and latter must rephrase
✓    We appreciate for the reviewer's reminding, the location of ZHD and ZWD has been exchanged.

157 In the procedure of horizontal voxel division, an approach is developed which enables the determination
✓    Thanks for the reviewer's reminding, this expression has been revised.

177 Further to the conclusion above it can also be concluded
✓    Thanks for the reviewer's reminding, this expression has been revised.

178 for the entire region using two/three/four-GNSS observations both increase with the
✓    Thanks for the reviewer's reminding, this expression has been revised.

195 following analysis focuses on: (1) investigating of two schemes in
✓    Thanks for the reviewer's reminding, this expression has been revised.

204 difference of voxels crossed by rays between Schemes 2 and 1 is not as expected for the case of
✓    Thanks for the reviewer's reminding, this expression has been revised.

219 It should be noted that the number of Galileo satellite is lower
✓    Thanks for the reviewer's reminding, this expression has been revised.

223 the highest
✓    Thanks for the reviewer's reminding, the word 'highest' has been used here.

225 only by about 3% more than
✓    Thanks for the reviewer's reminding, the word 'by' has been added.

226 of voxels for the three Schemes
✓    Thanks for the reviewer's reminding, this expression has been revised.

288 lower than that

✓ Thanks for the reviewer's reminding, the word 'smaller' has been replaced by 'lower'.

292 Hence it was
✓ Thanks for the reviewer's reminding, this word has been corrected.

323 an iterative procedure
✓ Thanks for the reviewer's question, the term 'produce' has been replaced by 'procedure'.

379 The upcoming full operability of the multi-constellation GNSS, is expected
✓ Thanks for the reviewer's suggestion, this expression has been revised.

381 results is not as expected.
✓ Thanks for the reviewer's reminding, this expression has been revised.

We appreciate for two reviewers' warm work earnestly, which has a significant improvement for our manuscript. And we hope that our corrections meet with the reviewers' requirements. Once again, thank you very much for your comments and suggestions.

[revised manuscript text omitted]
} A \\ H \\ V \end{pmatrix} \cdot x = \begin{pmatrix} y_{swd} \\ 0 \\ y_{rs} \end{pmatrix} \tag{4}$$

Where $H$ represents to the horizontal coefficient matrices while $V$ refers to the vertical coefficient matrices, respectively. $y_{swd}$ is a vector with SWD values while $y_{rs}$ is the *a priori* information obtained from the radiosonde information. The form of solution of the unknown wet refractivity vector can be written as:

$$\hat{x} = (A^T \cdot P_A \cdot A + H^T \cdot P_H \cdot H + V^T \cdot P_V \cdot V)^{-1} \cdot (A^T \cdot P_A \cdot y_{swd} + V^T \cdot P_V \cdot y_{rs}) \tag{5}$$

[revised manuscript text omitted]